# The potential shared role of inflammation in insulin resistance and schizophrenia: A bidirectional two-sample mendelian randomization study

Benjamin I. Perry[1,2]*, Stephen Burgess[3], Hannah J. Jones[4,5], Stan Zammit[4,5,6], Rachel Upthegrove[7], Amy M. Mason[8], Felix R. Day[9], Claudia Langenberg[9], Nicholas J. Wareham[9], Peter B. Jones[1,2], Golam M. Khandaker[1,2]

1 Department of Psychiatry, University of Cambridge School of Clinical Medicine, Cambridge, England, 2 Cambridgeshire and Peterborough NHS Foundation Trust, Cambridge, England, 3 MRC Biostatistics Unit, University of Cambridge, Cambridge, England, 4 NIHR Biomedical Research Centre, University Hospitals Bristol NHS Foundation Trust and University of Bristol, Bristol, United Kingdom, 5 Centre for Academic Mental Health, Population Health Sciences, Bristol Medical School, University of Bristol, Bristol, England, 6 MRC Centre for Neuropsychiatric Genetics and Genomics, Cardiff University, Cardiff, Wales, 7 Institute for Mental Health, University of Birmingham, Birmingham, England, 8 Department of Public Health and Primary Care, University of Cambridge, Cambridge, England, 9 MRC Epidemiology Unit, University of Cambridge School of Clinical Medicine, Cambridge, England

* bip20@medschl.cam.ac.uk

**Data Availability Statement:** The summary data for the 53 insulin resistance SNPs are available in the supporting information of Lotta et al [11] (10.

## Abstract

### Background

Insulin resistance predisposes to cardiometabolic disorders, which are commonly comorbid with schizophrenia and are key contributors to the significant excess mortality in schizophrenia. Mechanisms for the comorbidity remain unclear, but observational studies have implicated inflammation in both schizophrenia and cardiometabolic disorders separately. We aimed to examine whether there is genetic evidence that insulin resistance and 7 related cardiometabolic traits may be causally associated with schizophrenia, and whether evidence supports inflammation as a common mechanism for cardiometabolic disorders and schizophrenia.

### Methods and findings

We used summary data from genome-wide association studies of mostly European adults from large consortia (Meta-Analyses of Glucose and Insulin-related traits Consortium (MAGIC) featuring up to 108,557 participants; Diabetes Genetics Replication And Meta-analysis (DIAGRAM) featuring up to 435,387 participants; Global Lipids Genetics Consortium (GLGC) featuring up to 173,082 participants; Genetic Investigation of Anthropometric Traits (GIANT) featuring up to 339,224 participants; Psychiatric Genomics Consortium (PGC) featuring up to 105,318 participants; and Cohorts for Heart and Aging Research in Genomic Epidemiology (CHARGE) consortium featuring up to 204,402 participants). We conducted two-sample uni- and multivariable mendelian randomization (MR) analysis to

1038/ng.3714). Full summary statistics for all traits used in the primary analysis are freely and publicly available for download at consortia/group websites. Specifically; for fasting insulin, FPG, HbA1C and glucose tolerance summary data, see https://www.magicinvestigators.org/downloads/; For HDL, LDL and triglycerides summary data, see http://csg.sph.umich.edu/willer/public/lipids2013/; For BMI summary data, see https://portals.broadinstitute.org/collaboration/giant/index.php/GIANT_consortium_data_file s; For T2DM, see https://diagram-consortium.org/downloads.html; For leptin summary data, see ftp://ftp.ebi.ac.uk/pub/databases/gwas/summary_statistics/KilpelainenTO_26833098_GCST0 03368; For schizophrenia summary data, see https://www.med.unc.edu/pgc/download-results/. Summary GWAS data for CRP, which formed part of our post-hoc sensitivity analysis, are also publicly available from the primary GWAS study [35], and inquiries regarding use of CRP summary data can be sent to s.ligthart@erasmusmc.nl.

**Funding:** This report is independent research supported by the National Institute for Health Research (NIHR Doctoral Research Fellowship, BIP, DRF-2018-11-ST2-018). The views expressed in this publication are those of the author(s) and not necessarily those of the NHS, the National Institute for Health Research or the Department of Health and Social Care. GMK acknowledges funding support from the Wellcome Trust (Intermediate Clinical Fellowship; grant code: 201486/Z/16/Z), MQ: Transforming Mental Health (Data Science Award; grant code: MQDS17/40), and the Medical Research Council (MICA: Mental Health Data Pathfinder; grant code: MC_PC_17213). SZ & HJ are supported by the NIHR Biomedical Research Centre, University Hospitals Bristol NHS Foundation Trust and University of Bristol. SB is supported by a Sir Henry Dale Fellowship jointly funded by the Wellcome Trust and the Royal Society (Grant Number 204623/Z/16/Z). PBJ acknowledges funding from the MRC and MQ (as above), programmatic funding from NIHR (RP-PG- 0616-20003) and support from the Applied Research Collaboration East of England. RU acknowledges funding support from the NIHR (HTA grant code): 127700 and MRC (Therapeutic Target Validation in Mental Health grant code: MR/S037675/1). The funders had no role in study design, data collection and analysis, decision to publish, or preparation of the manuscript.

**Competing interests:** We have read the journal's policy and the authors of this manuscript have the following competing interests. SB is a paid

test whether (i) 10 cardiometabolic traits (fasting insulin, high-density lipoprotein and triglycerides representing an insulin resistance phenotype, and 7 related cardiometabolic traits: low-density lipoprotein, fasting plasma glucose, glycated haemoglobin, leptin, body mass index, glucose tolerance, and type 2 diabetes) could be causally associated with schizophrenia; and (ii) inflammation could be a shared mechanism for these phenotypes. We conducted a detailed set of sensitivity analyses to test the assumptions for a valid MR analysis. We did not find statistically significant evidence in support of a causal relationship between cardiometabolic traits and schizophrenia, or vice versa. However, we report that a genetically predicted inflammation-related insulin resistance phenotype (raised fasting insulin (raised fasting insulin (Wald ratio OR = 2.95, 95% C.I, 1.38–6.34, Holm-Bonferroni corrected $p$-value ($p$) = 0.035) and lower high-density lipoprotein (Wald ratio OR = 0.55, 95% C.I., 0.36–0.84; $p$ = 0.035)) was associated with schizophrenia. Evidence for these associations attenuated to the null in multivariable MR analyses after adjusting for C-reactive protein, an archetypal inflammatory marker: (fasting insulin Wald ratio OR = 1.02, 95% C.I, 0.37–2.78, $p$ = 0.975), high-density lipoprotein (Wald ratio OR = 1.00, 95% C.I., 0.85–1.16; $p$ = 0.849), suggesting that the associations could be fully explained by inflammation. One potential limitation of the study is that the full range of gene products from the genetic variants we used as proxies for the exposures is unknown, and so we are unable to comment on potential biological mechanisms of association other than inflammation, which may also be relevant.

## Conclusions

Our findings support a role for inflammation as a common cause for insulin resistance and schizophrenia, which may at least partly explain why the traits commonly co-occur in clinical practice. Inflammation and immune pathways may represent novel therapeutic targets for the prevention or treatment of schizophrenia and comorbid insulin resistance. Future work is needed to understand how inflammation may contribute to the risk of schizophrenia and insulin resistance.

## Author summary

### Why was this study done?

- Cardiometabolic disorders such as diabetes are up to 30% more common in people with schizophrenia than in the general population, and are among the predominant causes of a 10- to 15-year shortened life expectancy in people with schizophrenia.

- Insulin resistance, a precursor to diabetes, is sometimes detectable in young adults suffering their first episode of psychosis, which suggests that chronic lifestyle and clinical factors, such as smoking, physical inactivity, and medication side effects may not fully explain the comorbidity.

- Inflammation has been consistently associated with schizophrenia and cardiometabolic disorders, and so could be a common mechanism for schizophrenia and cardiometabolic disorders. This could help to at least in part explain why people who have

statistical consultant on PLOS Medicine's statistical board. CL is an Academic Editor on PLOS Medicine's editorial board. PBJ has received honoraria for providing scientific advice to Jansen, Ricordati and Lundbeck.

**Abbreviations:** BMI, body mass index; CHARGE, Cohorts for Heart and Aging Research in Genomic Epidemiology; CI, confidence interval; CRP, C-reactive protein; DAGs, directed acyclic graphs; CVD, cardiovascular disease; DIAGRAM, Diabetes Genetics Replication And Meta-analysis; FEP, first-episode psychosis; FPG, fasting plasma glucose; GIANT, Genetic Investigation of Anthropometric Traits; GLGC, Global Lipids Genetics Consortium; GWAS, genome-wide association study; HOMA, homeostasis model assessment; HbA1C, glycated haemoglobin; HDL, high-density lipoprotein; IVW, inverse variance weighted; LD, linkage disequilibrium; LDL, low-density lipoprotein; MAGIC, Meta-Analyses of Glucose and Insulin-related traits Consortium; MR, mendelian randomization; MR-PRESSO, MR pleiotropy residual sum and outlier; MVMR, multivariable MR; NLR, neutrophil to lymphocyte ratio; OR, odds ratio; PGC, Psychiatric Genomics Consortium; SD, standard deviation; SEs, standard errors; SNPs, single nucleotide polymorphisms; STROBE, Strengthening the Reporting of Observational Studies in Epidemiology; STROBE-MR, Strengthening the Reporting of MR studies; T2DM, type 2 diabetes mellitus.

schizophrenia also have higher rates of cardiometabolic disorders, over and above the commonly attributed lifestyle/clinical factors.

## What did the researchers do and find?

- To examine whether insulin resistance and 7 related cardiometabolic traits causally influence schizophrenia risk or vice versa, we conducted bidirectional, two-sample, uni- and multivariable mendelian randomizsation (MR) analyses. The MR approach uses genetic variants as proxies for modifiable exposures to untangle the problems of reverse causation and unmeasured confounding.

- To test a hypothesis that inflammation may be a common mechanism for schizophrenia and cardiometabolic disorders, we also examined a subset of genetic variants which were associated with inflammation as well as the cardiometabolic trait. We also used multivariable MR (MVMR) as a sensitivity analysis to adjust for C-reactive protein (CRP), an archetypal inflammatory marker, as a general downstream marker of systemic inflammation.

- After correction for multiple testing, overall, there was no significant evidence in support of a causal relationship between cardiometabolic traits and schizophrenia, or vice versa. However, we found evidence that supports a causal relationship of an inflammation-related insulin resistance phenotype with schizophrenia.

- Evidence for the association of an inflammation-related insulin resistance phenotype with schizophrenia attenuated fully in MVMR analysis after adjusting for CRP, suggesting that these associations may be underpinned by inflammation.

## What do these findings mean?

- These results suggest that cardiometabolic traits are unlikely to have a causal role in the pathogenesis of schizophrenia or vice versa. However, our results suggest that inflammation is related to the risk of both schizophrenia and insulin resistance, which may at least partly explain why they commonly occur in clinical practice.

- Treating or preventing inflammation may be a putative therapeutic option for prevention and/or treatment of both schizophrenia and comorbid insulin resistance.

- In the future, more research is needed to understand the biological mechanisms underpinning how inflammation may increase the risk of schizophrenia and insulin resistance.

## Introduction

Schizophrenia is a complex behavioural and cognitive syndrome characterised primarily by disruptions to perception and cognition [1]. It has a lifetime prevalence of around 0.4% [2] but carries a significant global disease burden [3]. Cardiometabolic disorders are up to 30% more common in schizophrenia than the general population [4] and are the leading contributors to premature death in these patients [5]. Their increased prevalence in schizophrenia is

commonly attributed to the adverse effects of antipsychotic medications [6] or lifestyle factors such as physical inactivity and a poor diet [7], but this is unlikely to be the whole story. While the aforementioned factors contribute cumulative risk over time [8], recent meta-analyses of case–control studies suggest that a phenotype of raised fasting insulin, raised triglycerides, and low high-density lipoprotein (HDL) cholesterol, indicative of insulin resistance [9–11], is associated with relatively young antipsychotic-naïve patients with first-episode psychosis (FEP) [12,13], and, cross-sectionally, with psychotic symptoms in young adults [14]. Therefore, insulin resistance, which is a significant risk factor for type 2 diabetes mellitus (T2DM) and obesity, might be causally related to, or share pathophysiologic mechanisms with schizophrenia.

The majority of existing research in the field is cross-sectional, and therefore cannot confirm whether cardiometabolic disorders are a cause or consequence of illness (i.e., reverse causality). For example, 1 longitudinal study found no evidence for an association between insulin resistance in childhood and risk of psychosis in late adolescence [14]. Additionally, while previous studies have adjusted for a number of potential confounders, residual confounding, which is a limitation of both cross-sectional and longitudinal research, could still be relevant. Mendelian randomization (MR) analysis can address these limitations by using genetic variants inherited randomly at conception as unconfounded proxies of a modifiable exposure, to examine whether the exposure may have a causal effect on a disease outcome [15]. MR studies of cardiometabolic traits and schizophrenia are limited, have focused on a very limited set of cardiometabolic exposures, and have reported mixed findings [16,17]. To our knowledge, MR studies examining associations between a wide range of cardiometabolic traits and schizophrenia are lacking. Such studies may help to identify common potentially causal risk factors and pathophysiologic mechanisms for these physical and psychiatric illnesses.

Inflammation could be pathophysiologically related to cardiometabolic disorders and schizophrenia. Higher levels of circulating inflammatory markers have been associated with both psychosis and cardiometabolic disorders, both cross-sectionally and longitudinally [18–20]. MR studies have reported potential causal associations between inflammation, particularly C-reactive protein (CRP) and interleukin-6 (IL-6), and schizophrenia [21,22]. CRP and IL-6 are also implicated in pathogenesis of insulin resistance [23] and may exaggerate the effects of insulin resistance on psychosis risk in young adults [14]. However, to our knowledge, no MR studies have examined whether inflammation could be pathophysiologically related to insulin resistance and schizophrenia, for example, via mediating or common causal mechanisms.

Therefore, we have conducted a study to examine evidence in support of 4 scenarios regarding the potential relationships between inflammation, insulin resistance, and schizophrenia: (1) Inflammation is a common cause (confounder) between insulin resistance and schizophrenia; (2) insulin resistance mediates an association between inflammation and schizophrenia; or vice versa; (3) inflammation is a common cause (confounder) between schizophrenia and insulin resistance; and (4) schizophrenia mediates an association between inflammation and insulin resistance. See S1 Methods for directed acyclic graphs (DAGs) illustrating the proposed mechanisms.

First, we carried out MR analyses to test whether 10 cardiometabolic traits related to insulin resistance (fasting insulin, triglycerides, HDL, low-density lipoprotein (LDL), fasting plasma glucose (FPG), body mass index (BMI), glucose tolerance, leptin, glycated haemoglobin (HbA1C), and T2DM) could be causally associated with schizophrenia. To test the direction of association, we used genetically predicted levels of cardiometabolic traits as exposures and schizophrenia as the outcome and vice versa. Next, we examined whether inflammation could be a shared mechanism linking insulin resistance and schizophrenia using MR analyses including genetic variants for each cardiometabolic trait that were also associated with a marker of inflammation. Finally, we used multivariable MR (MVMR) analysis to control for

genetic associations of cardiometabolic traits with CRP, an archetypal general inflammatory marker, which we used as a general measure for systemic inflammation.

## Methods

### Selection of genetic variants related to cardiometabolic traits and schizophrenia

For fasting insulin, triglycerides, and HDL, we used a set of 53 single nucleotide polymorphisms (SNPs) reported to be associated with all 3 traits, representative of an insulin resistance phenotype, from a recent meta genome-wide association study (GWAS) of 188,577 European adults, which adjusted for BMI [11]. In our study, we included SNPs reaching genome-wide significance for the corresponding trait. Summary statistics for genome-wide significant SNPs were also obtained for 6 related continuous (FPG, HbA1C, LDL, BMI, leptin, and glucose tolerance) and 1 binary (T2DM) cardiometabolic traits from recent large GWAS (S2–S10 Methods). We obtained summary statistics for schizophrenia from a recent GWAS from the Psychiatric Genomics Consortium (PGC) [24] based on 40,675 cases and 64,643 European controls. The degree of sample overlap between exposure and outcome samples was likely to be low since exposure and outcome data were obtained from different consortia [25].

### Ethics statement

Our study was a secondary analysis of the above publicly available data. Informed consent was sought for all participants per the original GWAS protocols, and all ethical approvals for the GWAS were obtained by original GWAS authors.

### Statistical analysis

The analysis plan was prospectively conceived by the authors in 2019 but was not formally deposited in a repository or database. All described analyses were planned *a priori* except for the following: a) the analysis of inflammation-related SNPs at a less-stringent significance threshold (see the 'Analysis using inflammation-related SNPs' section below); b) the MVMR analysis including CRP (see the 'Adjustment for Inflammation' section below). These analyses were conceived and conducted in light of findings from the primary analysis, to further probe whether inflammation could explain the results. We obtained summary-level data (SNP rs number, β-coefficient or log odds ratio (OR), standard errors or 95% confidence intervals (CIs), effect allele, other allele, *p*-value, effect allele frequency, sample size, and number of cases/controls) from each GWAS. Where a specific instrument SNP was not available in the outcome dataset, we located proxy SNPs using linkage disequilibrium (LD) tagging ($r^2 > 0.8$) via LDlink [26]. Alleles were harmonised based on matching alleles, and the resulting instruments were clumped for LD to ensure independence (10,000 kb pairs apart, $r^2 < 0.001$). In the event of palindromic SNPs, the forward strand was inferred where possible using allele frequency information. We performed bidirectional analysis (i.e., with schizophrenia as exposure and cardiometabolic traits as outcomes) to examine direction of association. Statistical analysis was conducted using the TwoSampleMR package (v0.5.4) [27] for R (The R Foundation for Statistical Computing, Vienna, Austria) [28]. Our primary MR analysis method was inverse variance weighted (IVW) regression where at least two exposure SNPs were available for analysis. Where one exposure SNP was available for analysis, we used the Wald ratio method. We also conducted weighted median and MR–Egger regression analysis (S11 Methods). For the binary outcome of schizophrenia, the estimates for continuous exposures (fasting insulin, HDL, triglycerides, LDL, FPG, BMI, HbA1C, glucose tolerance, and leptin) represent log-odds

ratios converted into ORs, representing the increase in risk of schizophrenia per standard deviation (SD) of exposure, and 95% CIs. For binary exposures (T2DM), the estimates represent the OR for schizophrenia per unit increase in the log-odds of T2DM. For continuous cardiometabolic outcomes, β-coefficients represent the SD increase in exposure per unit increase in the log-odds of schizophrenia, with standard errors (SEs).

We performed several sensitivity analyses to check the validity of our results. Heterogeneity among SNPs included in each analysis was examined using the Cochran Q test. We checked for horizontal pleiotropy using the MR–Egger regression intercept alongside a more recent and robust method to detect horizontal pleiotropy and outliers, "MR pleiotropy residual sum and outlier" (MR-PRESSO) [29]. Using MR-PRESSO, we used the global test to examine for horizontal pleiotropy, and where evident, used the method to correct the IVW-estimate via outlier removal (S11 Methods). We examined for measurement error in SNP-exposure associations using the $I^2_{GX}$ statistic [30]. This study is reported as per the Strengthening the Reporting of MR studies (STROBE-MR) guideline [31] (S1 Checklist) and the Strengthening the Reporting of Observational Studies in Epidemiology (STROBE) statement [32] (S2 Checklist).

### Analysis using inflammation-related SNPs

Next, we repeated MR analysis using only inflammation-related SNPs for each cardiometabolic risk factor as an instrumental variable for the outcome of schizophrenia. We did this to test the hypothesis that these SNPs may represent a mechanism involving inflammation. This could be via, for example, a common causal basis (panel A in S1 Methods) or via vertical (mediating) pleiotropy [27] (panel B in S1 Methods). We used Phenoscanner v2 (University of Cambridge, United Kingdom) [33] to examine each SNP associated with each cardiometabolic risk factor, to identify SNPs that were also associated with a measure of inflammation, defined as blood concentration/count of cytokines (such as chemokines, interferons, interleukins, lymphokines, or tumour necrosis factors), acute phase proteins (e.g., CRP), or immune cells (e.g., neutrophils and lymphocytes). Primarily, we considered inflammation-related SNPs at genome-wide significance ($p<5\times10^{-8}$) to maximise specificity. We also performed a sensitivity analysis by including inflammation-related SNPs at a less-stringent nominal significance threshold ($p<1\times10^{-4}$) to increase sensitivity to inflammation-related SNPs [34] (S12–S17 Methods).

Using the same method, we identified inflammation-related schizophrenia SNPs (S18 Methods) and used them as instrumental variables in MR analysis examining cardiometabolic traits as outcomes.

### Adjustment for inflammation

As a post hoc sensitivity analysis to estimate whether any associations evident above may be explained by inflammation, we carried out MVMR analysis [34,35] using the 53 SNPs for fasting insulin, triglycerides, and HDL, representative of an insulin resistance phenotype, as exposures with schizophrenia as the outcome, after conditioning on the associations of these 53 SNPs with CRP. We chose CRP because it is a widely used downstream measure of systemic inflammation, and publicly available data from large-scale GWAS for CRP are available. Summary statistics for CRP were obtained from a recent large GWAS based on 204,402 participants [36]. For CRP as an exposure in MVMR, we used all independent (10,000 kb pairs apart, $r^2 < 0.001$) SNPs reported to be conditionally associated with CRP and located within the *CRP* coding region (S19 Methods).

### Correction for multiple testing

Statistical significance was estimated using the Holm–Bonferroni correction method [37], correcting for the number of exposures tested at each stage of analysis.

## Results

### MR analyses using all genetic variants associated with insulin resistance and other cardiometabolic traits

We did not find significant evidence for associations between genetically-predicted levels of cardiometabolic traits and schizophrenia, using the primary IVW analysis method. Evidence using the weighted median method for associations between genetically-predicted levels of triglycerides (weighted median OR = 1.26; 95% C.I., 1.06–1.50; corrected $p$ = 0.090) and HDL (weighted median OR = 0.79; 95% C.I., 0.65–0.95; corrected $p$ = 0.126) with schizophrenia did not survive correction for multiple testing (Table 1).

**Table 1. MR analyses of cardiometabolic traits and schizophrenia using all SNPs.**

| Risk Factor | SNPs, No.[a] | Method | Odds Ratio (95% C.I.) | $p$-value | Corrected $p$-value[b] |
|---|---|---|---|---|---|
| Fasting Insulin | 9 | IVW | 1.13 (0.76–1.70) | 0.548 | 1.000 |
| | | Weighted Median | 0.98 (0.68–1.41) | 0.920 | 1.000 |
| | | MR Egger | 9.24 (1.82–46.97) | 0.028 | 0.280 |
| Triglycerides | 9 | IVW | 1.16 (0.86–1.56) | 0.334 | 1.000 |
| | | Weighted Median | 1.26 (1.06–1.50) | 0.009 | 0.090 |
| | | MR Egger | 1.31 (0.84–2.03) | 0.308 | 1.000 |
| HDL | 14 | IVW | 0.94 (0.71–1.23) | 0.649 | 1.000 |
| | | Weighted Median | 0.79 (0.65–0.95) | 0.010 | 0.126 |
| | | MR Egger | 0.67 (0.45–0.99) | 0.067 | 0.670 |
| Fasting Plasma Glucose | 18 | IVW | 1.07 (0.87–1.31) | 0.522 | 1.000 |
| | | Weighted Median | 1.01 (0.84–1.23) | 0.887 | 1.000 |
| | | MR Egger | 1.13 (0.74–1.74) | 0.584 | 1.000 |
| Type 2 Diabetes Mellitus | 27 | IVW | 0.93 (0.78–1.12) | 0.470 | 1.000 |
| | | Weighted Median | 0.93 (0.80–1.09) | 0.375 | 1.000 |
| | | MR Egger | 1.03 (0.66–1.62) | 0.895 | 1.000 |
| Body Mass Index | 81 | IVW | 1.05 (0.89–1.24) | 0.554 | 1.000 |
| | | Weighted Median | 1.07 (0.92–1.24) | 0.383 | 1.000 |
| | | MR Egger | 1.43 (0.97–2.10) | 0.103 | 1.000 |
| HbA1C | 36 | IVW | 1.01 (0.76–1.32) | 0.956 | 1.000 |
| | | Weighted Median | 1.12 (0.82–1.51) | 0.483 | 1.000 |
| | | MR Egger | 1.33 (0.79–2.23) | 0.295 | 1.000 |
| Glucose Tolerance | 7 | IVW | 0.98 (0.85–1.14) | 0.800 | 1.000 |
| | | Weighted Median | 1.10 (0.87–1.15) | 0.993 | 1.000 |
| | | MR Egger | 1.85 (0.95–3.32) | 0.094 | 0.940 |
| LDL | 74 | IVW | 0.99 (0.93–1.05) | 0.679 | 1.000 |
| | | Weighted Median | 0.97 (0.90–1.03) | 0.322 | 1.000 |
| | | MR Egger | 0.98 (0.90–1.07) | 0.692 | 1.000 |
| Leptin | 4 | IVW | 1.97 (0.90–4.31) | 0.091 | 0.910 |
| | | Weighted Median | 1.18 (0.66–2.11) | 0.579 | 1.000 |
| | | MR Egger | 3.29 (0.56–17.22) | 0.358 | 1.000 |

HDL = high-density lipoprotein; HbA1C = glycated haemoglobin; LDL = low-density lipoprotein; IVW = inverse variance weighted regression; SNPs = single nucleotide polymorphisms

[a]Number of SNPs remaining after clumping for independence

[b] Each analysis method (IVW, Weighted Median and MR Egger) corrected using the Holm-Bonferroni method for 10 cardiometabolic markers

Estimates represent ORs for schizophrenia per SD increase in exposure (per unit-increase in log-odds of exposure for T2DM).

### MR analyses using inflammation-related genetic variants for insulin resistance and other cardiometabolic traits

After testing only genome-wide significant inflammation-related SNPs for cardiometabolic traits, we found evidence for associations of inflammation-related genetically-predicted fasting insulin (Wald Ratio OR = 2.95; 95% C.I., 1.38–6.34; corrected $p$ = 0.035) and HDL (Wald Ratio OR = 0.55; 95% CI, 0.36–0.84; corrected $p$ = 0.035) with schizophrenia. We could not include any genome-wide significant inflammation-related variants for triglycerides, leptin or glucose tolerance. In our sensitivity analysis featuring inflammation-related cardiometabolic variants at a less stringent significance threshold, evidence persisted for associations of inflammation-related genetically-predicted fasting insulin (IVW OR = 1.74; 95% C.I., 1.08–2.98; corrected $p$ = 0.030) and HDL (IVW OR = 0.78; 95% C.I., 0.62–0.92; corrected $p$ = 0.036) with schizophrenia. In addition, we found evidence for an association of genetically-predicted inflammation-related triglycerides (IVW OR = 1.24; 95% C.I., 1.07–1.55; corrected $p$ = 0.036) with schizophrenia (Table 2; Fig 1 & Fig 2).

### Adjustment for inflammation

MVMR analysis for inflammation-related SNPs of fasting insulin, triglycerides, and HDL with schizophrenia showed that the univariable associations fully attenuated after controlling for the genetic associations of these variants with CRP, in analyses involving both inflammation-related SNPs at genome-wide and nominal significance levels. Controlling for CRP had negligible effect on MR estimates based on all genetic variants (Fig 3, S1 and S2 Results).

### Test for bidirectionality using schizophrenia as exposure

We did not find statistically significant MR associations between schizophrenia and any cardiometabolic trait after correction for multiple testing (S3 Results, S1 Fig). Similarly, we did not find statistically significant MR associations of inflammation-related schizophrenia variants with cardiometabolic traits after correction for multiple testing (S4 Results, S1 Fig).

### Test for horizontal pleiotropy

Using the MR-Egger regression intercept test, we found evidence of potential horizontal pleiotropy for BMI and HDL in the all-SNP analysis, but no evidence for horizontal pleiotropy for any cardiometabolic exposure in the inflammation-related SNP analysis. Using MR-PRESSO however, we found evidence that horizontal pleiotropy was likely to have affected estimates for all cardiometabolic exposures in the all-SNP analysis ($p$ value for global test all ≤0.020), and both LDL and T2DM in the inflammation-related SNP analysis. Following MR-PRESSO outlier correction, evidence strengthened for the association of triglycerides with schizophrenia in the all-SNP analysis (MR-PRESSO IVW β = 0.23, S.E. 0.06, $p$ = 0.008), but outlier-corrected IVW estimates for other exposures were not significantly altered.

In the bidirectional analyses, both MR-PRESSO and the MR-Egger regression intercept suggested horizontal pleiotropy affecting the outcomes of HDL, BMI and LDL (all $p$<0.05). Following outlier correction, there was evidence for a weak protective effect of schizophrenia on BMI (β = -0.04, S.E. 0.02, $p$ = 0.014). MR-PRESSO additionally revealed possible horizontal pleiotropy affecting the outcomes of fasting insulin, triglycerides and T2DM (p for MR-PRESSO global test all <0.05) (S5–S12 Results), but outlier-corrected IVW estimates were not significantly altered.

### Test for heterogeneity of instruments

In the analyses based on all SNPs, the majority of cardiometabolic traits demonstrated evidence of heterogeneity, which was reduced in the inflammation-related SNP analysis (S5–S8

**Table 2.** MR analyses of inflammatory-related cardiometabolic SNPs and schizophrenia.

| Risk Factor | Method | Genome-Wide Significant Inflammatory-Related SNPs | | | | Nominally Significant Inflammatory-Related SNPs | | | |
| --- | --- | --- | --- | --- | --- | --- | --- | --- | --- |
| | | SNPs, No. | Odds Ratio (95% C.I.) | p-value | Corrected p-value[a] | SNPs, No. | Odds Ratio (95% C.I.) | p-value | Corrected p-value[a] |
| Fasting Insulin | IVW / Wald Ratio | 1 | 2.95 (1.38–6.34) | 0.005 | 0.035 | 5 | 1.74 (1.08–2.98) | 0.003 | 0.030 |
| | Weighted Median | | | | | | 1.40 (0.83–2.34) | 0.203 | 1.000 |
| | MR Egger | | | | | | 7.20 (1.03–50.54) | 0.141 | 0.987 |
| Triglycerides | IVW / Wald Ratio | 0 | * | * | * | 4 | 1.24 (1.07–1.55) | 0.004 | 0.036 |
| | Weighted Median | | | | | | 1.26 (1.06–1.50) | 0.009 | 0.063 |
| | MR Egger | | | | | | 1.29 (1.02–1.63) | 0.167 | 0.987 |
| HDL | IVW / Wald Ratio | 1 | 0.55 (0.36–0.84) | 0.005 | 0.035 | 7 | 0.78 (0.62–0.92) | 0.004 | 0.036 |
| | Weighted Median | | | | | | 0.77 (0.64–0.94) | 0.008 | 0.056 |
| | MR Egger | | | | | | 0.68 (0.51–0.91) | 0.047 | 0.288 |
| Fasting Plasma Glucose | IVW | 2 | 1.53 (0.39–5.97) | 0.537 | 1.000 | 4 | 1.04 (0.36–2.98) | 0.945 | 1.000 |
| | Weighted Median | | | | | | 1.08 (0.63–1.86) | 0.776 | 1.000 |
| | MR Egger | | | | | | 8.44 (0.65–120.54) | 0.409 | 1.000 |
| Type 2 Diabetes Mellitus | IVW | 7 | 0.94 (0.59–1.48) | 0.776 | 1.000 | 10 | 0.97 (0.71–1.33) | 0.850 | 1.000 |
| | Weighted Median | | 1.05 (0.26–4.32) | 0.941 | 1.000 | | 1.05 (0.74–1.48) | 0.781 | 1.000 |
| | MR Egger | | 1.40 (0.32–6.08) | 0.668 | 1.000 | | 1.42 (0.59–3.38) | 0.458 | 1.000 |
| HbA1C | IVW | 7 | 1.20 (0.67–2.13) | 0.546 | 1.000 | 10 | 1.02 (0.64–1.61) | 0.942 | 1.000 |
| | Weighted Median | | 0.93 (0.46–1.85) | 0.832 | 1.000 | | 0.95 (0.54–1.69) | 0.865 | 1.000 |
| | MR Egger | | 1.68 (0.39–7.21) | 0.508 | 1.000 | | 1.18 (0.41–3.37) | 0.767 | 1.000 |
| Body Mass Index | IVW | 4 | 1.23 (0.88–1.71) | 0.229 | 1.000 | 12 | 1.48 (0.76–2.87) | 0.249 | 1.000 |
| | Weighted Median | | 1.15 (0.80–1.65) | 0.451 | 1.000 | | 1.16 (0.85–1.58) | 0.350 | 1.000 |
| | MR Egger | | 0.77 (0.33–1.79) | 0.650 | 1.000 | | 3.36 (0.61–18.45) | 0.399 | 1.000 |
| LDL | IVW | 13 | 0.96 (0.79–1.17) | 0.687 | 1.000 | 23 | 0.93 (0.79–1.10) | 0.420 | 1.000 |
| | Weighted Median | | 0.91 (0.80–1.04) | 0.181 | 1.000 | | 0.91 (0.80–1.04) | 0.129 | 0.987 |
| | MR Egger | | 0.81 (0.58–1.14) | 0.254 | 1.000 | | 0.82 (0.62–1.11) | 0.220 | 0.987 |
| Leptin | IVW | 0 | * | * | * | 2 | 1.56 (0.77–3.17) | 0.221 | 0.987 |
| Glucose Tolerance | IVW | 0 | * | * | * | 2 | 1.06 (0.82–1.56) | 0.882 | 1.000 |

HDL = high-density lipoprotein; HbA1C = glycated haemoglobin; LDL = low-density lipoprotein; IVW = inverse variance weighted regression; SNPs = single nucleotide polymorphisms

[a]Each analysis method (IVW, Weighted Median and MR Egger) corrected using the Holm-Bonferroni method

*no inflammatory-related SNPs includedEstimates represent ORs for schizophrenia per SD increase in exposure (or per unit-increase in log-odds of binary exposures e.g. T2DM).

Results). There was limited evidence of heterogeneity in the sensitivity analyses based on inflammation-related SNPs for T2DM, BMI, and HbA1C only.

## Test for measurement error

Results for the $I^2_{GX}$ tests for SNP-exposure associations revealed some evidence for potential measurement error, which may have biased MR–Egger analyses in the analyses with leptin, glucose tolerance, T2DM, and schizophrenia as exposures (S13 Results).

## A. Fasting Insulin

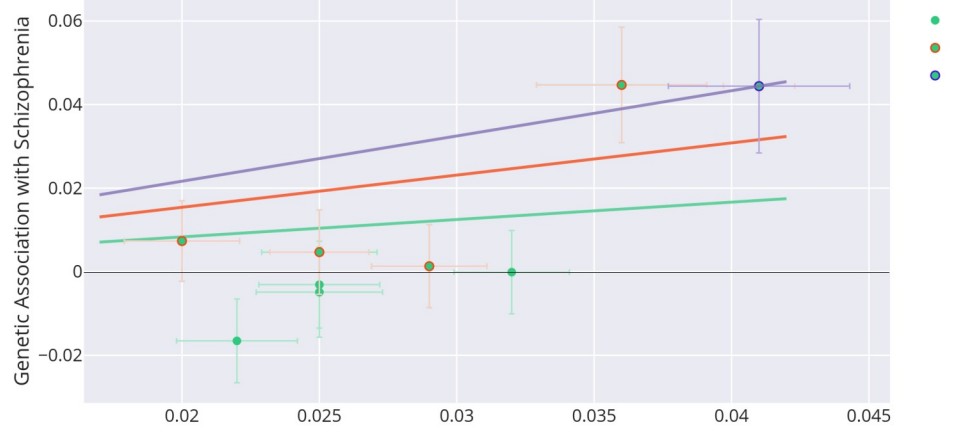

## B. Triglycerides

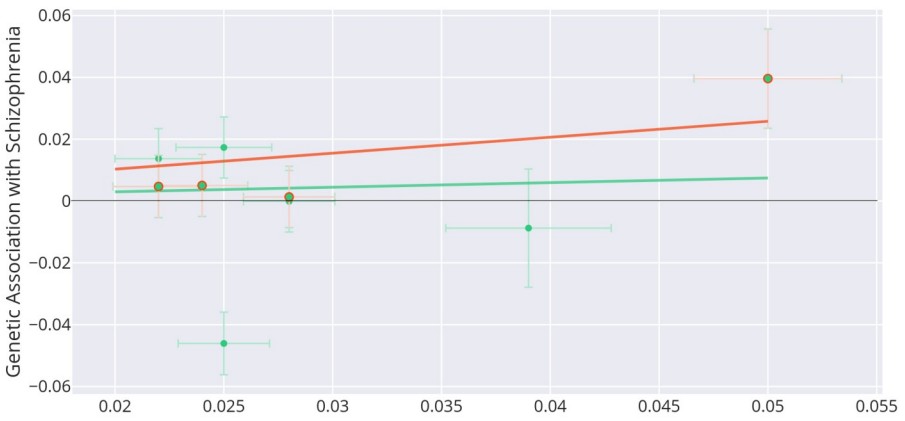

## C. HDL

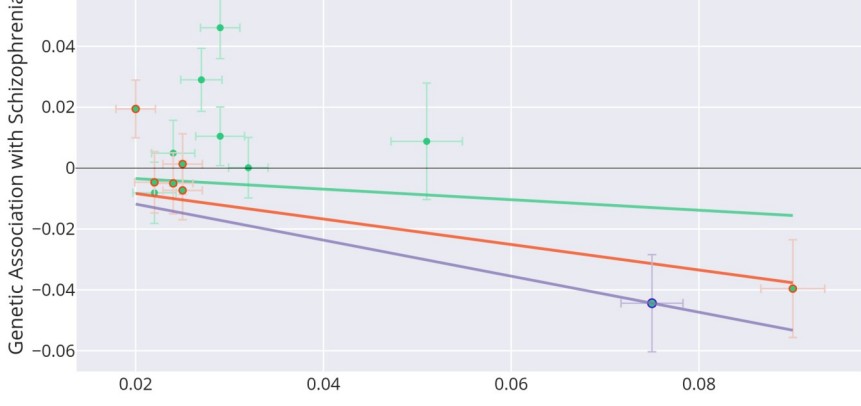

**Fig 1. MR Analyses Testing Associations between Insulin Resistance Phenotypes (Fasting Insulin (A), Triglycerides (B) and HDL (C)) and Schizophrenia, Highlighting Inflammation-Related SNPs.** Points in plots represent the association of the genome-wide significant insulin-resistance single nucleotide polymorphisms (SNPs) and their association with schizophrenia (Y axis) and the exposure (X axis). SNPs are denoted by green points in the plot. Inflammation-

related SNPs at genome-wide significance are denoted by a purple border. Inflammation-related SNPs at nominal significance are denoted by a red border. Whiskers represent standard errors. Lines on the plot represent inverse-variance weighted (>1 SNP) or linear regression (1 SNP) of all-SNPs (green line), inflammation-related SNPs at genome-wide significance (purple line) and inflammation-related SNPs at nominal significance (red line).

## Discussion

### Main findings

We conducted bidirectional uni- and multivariable two-sample MR analyses using large publicly available genomic datasets to first examine for associations that support a causal relationship between insulin resistance and related cardiometabolic traits and schizophrenia, and second, to examine whether there is evidence to support that inflammation may be a common causal mechanism for insulin resistance and schizophrenia. Using our primary IVW analysis method, we did not find evidence in support of a causal association between genetically-predicted cardiometabolic traits and schizophrenia. However, we found weak evidence using the weighted median method in support of a causal association of genetically-predicted levels of triglycerides and HDL with schizophrenia, but this association did not survive correction for multiple testing and the estimate may have been affected by horizontal pleiotropy. We found more consistent evidence for an association of the insulin resistance phenotype of fasting insulin, triglycerides, and HDL [11] with schizophrenia when we examined only genetic variants also associated with inflammation. Using two *p*-value cut-offs for inflammation-related SNPs, we found that the strength of association with schizophrenia increased as the specificity toward inflammation-related SNPs increased. In MVMR analyses adjusting for CRP, those estimates attenuated fully to the null. We found no evidence in bidirectional analyses in support of a causal relationship of schizophrenia with insulin resistance (panels C and D in S1 Methods). Together, our results are therefore most consistent with inflammation as a common cause for insulin resistance and schizophrenia (panel A in S1 Methods).

### Inflammation as a common cause for schizophrenia and insulin resistance

Three aspects of our results point towards inflammation as a common cause for insulin resistance and schizophrenia (panel A in S1 Methods). First, we did not find convincing overall evidence for a causal relationship between insulin resistance and schizophrenia (likely ruling out panel B in S1 Methods). Second, in our analyses of inflammation-related variants for the cardiometabolic traits, we found strong and consistent evidence in support of a potential causal relationship of fasting insulin, HDL and triglycerides with schizophrenia, and the strength of association with schizophrenia increased as the specificity toward inflammation-related SNPs increased. Third, we used MVMR to evidence that after controlling for CRP, an archetypal general inflammatory marker, the associations between inflammation-related genetic variants for insulin resistance and schizophrenia completely attenuated. This result suggests that the observed associations for the inflammation-related variants are at least in part explained by inflammation. Together, the results are consistent with the idea that inflammation may be a common causal mechanism for insulin resistance and schizophrenia.

Evidence for a common causal mechanism between insulin resistance and schizophrenia may help to explain why schizophrenia is associated with higher rates of insulin resistance even in early stages of illness, when the cumulative effects of medication and lifestyle factors are relatively small [12,38]. Anti-inflammatory agents, of which several have shown promise in treating the symptoms of schizophrenia [39], should therefore be considered as a putative therapeutic target for prevention and treatment of cardiometabolic disorders in schizophrenia.

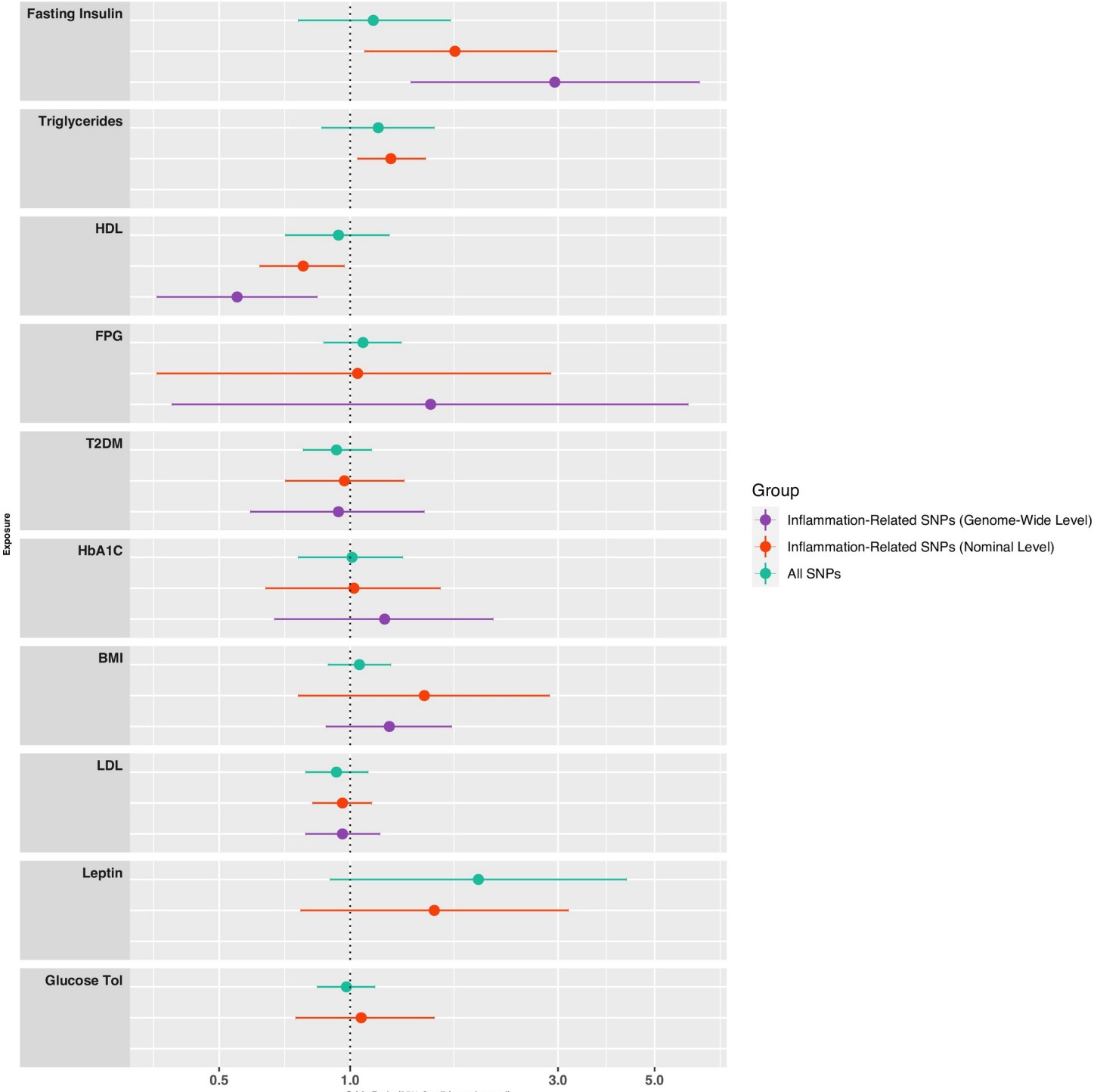

**Fig 2. MR analyses testing associations between cardiometabolic traits and schizophrenia.** Forest plot presents ORs and 95% CIs for associations between cardiometabolic traits and schizophrenia using IVW / Wald Ratio MR analyses based on all single nucleotide polymorphisms (SNPs) associated with each risk factor (green), inflammation-related SNPs at genome-wide significance (purple), and inflammation-related SNPs at nominal significance (red). See Tables 1 and 2 for the number of SNPs used in each analysis. HDL = High Density Lipoprotein; T2DM = Type 2 Diabetes Mellitus; BMI = Body Mass Index; FPG = Fasting Plasma Glucose; LDL = Low-Density Lipoprotein; HbA1C = Glycated Haemoglobin; Glucose Tol = Glucose Tolerance.

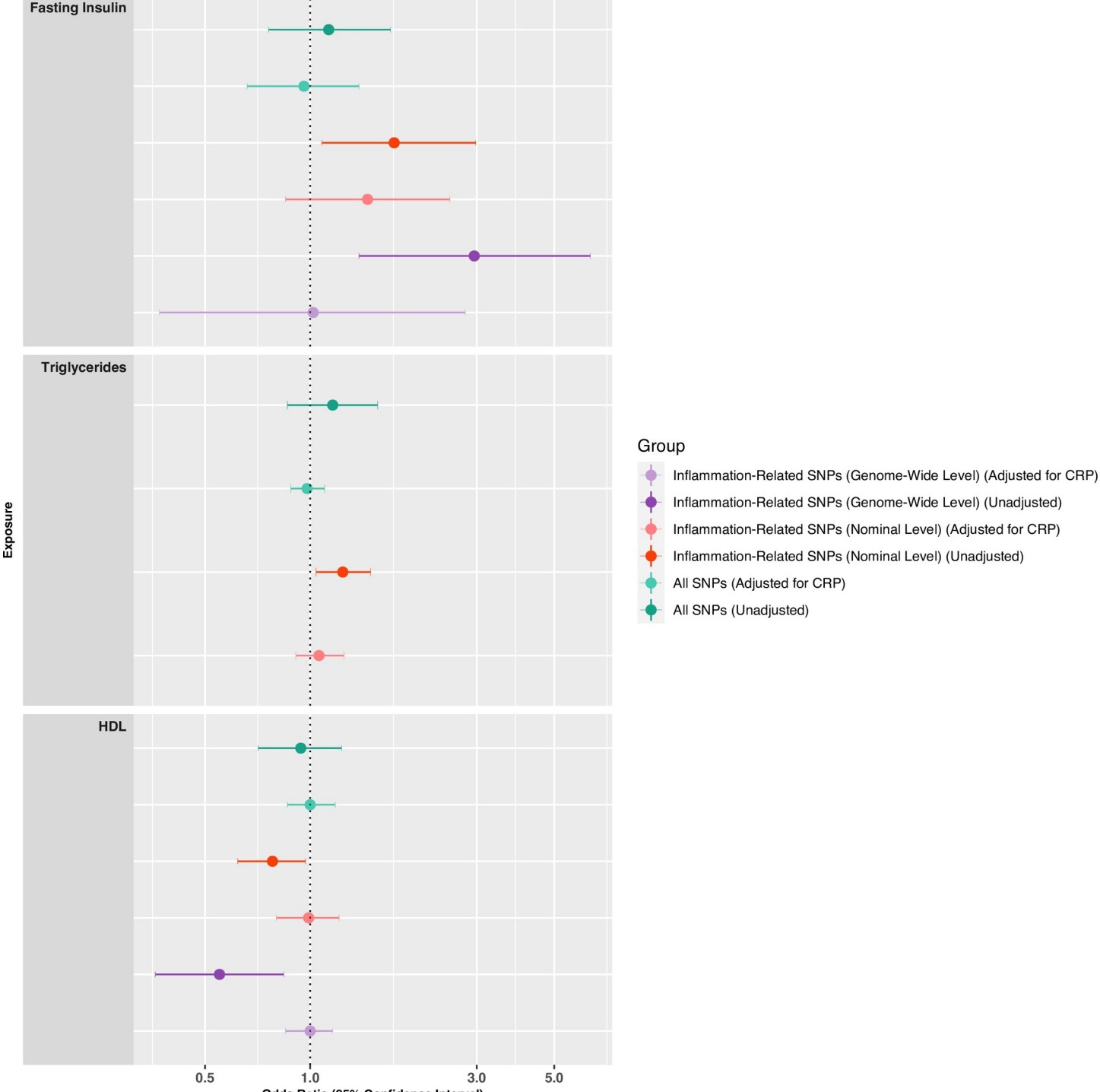

**Fig 3. Multivariable MR analysis testing associations between insulin resistance phenotypes and schizophrenia after controlling for genetic associations for CRP.**
Forest plot presents ORs and 95% CIs for inverse-variance weighted regression (IVW) MR associations between insulin resistance phenotypes and schizophrenia using all single nucleotide polymorphisms (SNPs) (dark green), and after controlling for association of these SNPs with C-reactive protein (CRP) using multivariable MR (MVMR) (light green). The forest plot also presents ORs and 95% CIs for IVW / Wald Ratio MR associations between insulin resistance phenotypes and schizophrenia using inflammation-related SNPs (genome-wide significance = dark purple; nominal significance = red), and after controlling for association of these SNPs with CRP using MVMR (genome-wide significance = light purple; nominal significance = light red). HDL = high-density lipoprotein.

We used CRP, an archetypal downstream inflammatory marker, as a means of gauging the effect of systemic inflammation in MVMR analysis, rather than hypothesising a specific role for CRP in the relationship between insulin resistance and schizophrenia. Nevertheless, CRP has observationally shown in both cross-sectional [40] and longitudinal [41] research to be associated with schizophrenia, although such findings are limited by the potential of residual confounding and reverse causality. Interestingly, however, MR findings have reported that genetically predicted CRP may have a protective effect on schizophrenia [21], with authors positing that a genetically attenuated ability to produce CRP may predispose to more insidious and chronic infections. In our MVMR analysis, attenuation of insulin resistance schizophrenia associations after controlling for CRP is consistent with inflammation being associated with both exposure and outcome, albeit "negatively" with the latter. Further research is needed to explore potential mechanisms of association between CRP and schizophrenia.

Many of the SNPs included in the inflammation-related analysis were associated with neutrophils and/or lymphocytes. A raised neutrophil to lymphocyte ratio (NLR) is a marker of systemic inflammation and is known to be associated with schizophrenia [42] and insulin resistance [43]. We were unable to find large GWAS studies conducted in European populations for NLR or for other inflammatory markers, which we might have used in MVMR analyses in place of CRP.

Based on our findings, we are unable to completely rule out the possibility that insulin resistance may mediate an inflammation–schizophrenia association (panel B in S1 Methods), since there was weak evidence that did not survive correction for multiple testing for an association of triglycerides and HDL with schizophrenia using the weighted median method, and in our MR-PRESSO sensitivity analysis, evidence from the outlier-corrected IVW analysis suggested a possible association between triglycerides and schizophrenia. These findings are broadly similar to 1 previous MR study [17], which reported only weak evidence of an association between the homeostasis model assessment (HOMA), a measure of insulin resistance, on schizophrenia. Another MR study [16] reported a genetic association between fasting insulin and schizophrenia, although the evidence attenuated after adjustment for BMI. To account for BMI, we obtained summary statistics for genetic variants related to insulin resistance after controlling for BMI [11]. The previous MR study included an ethnically heterogeneous sample, increasing the potential for population stratification bias. We used genetic data from a more ethnically homogenous GWAS of schizophrenia [24]. Nevertheless, while our results in the all-SNP analysis suggested weak evidence for triglycerides and HDL, which may reflect an insulin resistance phenotype, the evidence did not survive correction for multiple testing and requires replication in future when larger GWAS samples are available.

The implications of our findings with regard to shared causal mechanisms should not distract clinicians from focusing on the assessment and management of malleable lifestyle factors related to cardiometabolic disorders in people with schizophrenia. Factors such as poorer diet, reduced exercise and smoking, which are associated with schizophrenia [7,44,45], may predispose to an inflammatory state [46]. Therefore, it is possible that lifestyle factors exacerbate a feedback loop between inflammation, insulin resistance, and schizophrenia by increasing both inflammation and insulin resistance, eventually leading to T2DM and other cardiometabolic disorders such as obesity and cardiovascular disease (CVD). In addition to the potential therapeutic potential of anti-inflammatory medications, malleable lifestyle factors must continue to remain crucial targets [47,48] for the prevention of cardiometabolic morbidity in people with schizophrenia.

## Additional findings

We report that after outlier correction, schizophrenia had a weak protective effect on BMI. This finding complements estimates from previous research [53] using LD score regression,

though we are able to advance previous findings since genetic correlation analyses are unable to test direction of association. This finding suggests that weight gain associated with schizophrenia is unlikely to be a feature of the illness itself but could be attributed to iatrogenic or lifestyle effects. Moreover, the "lean insulin-resistance" phenotype may be associated with higher levels of inflammation [54] and warrants further research in the context of schizophrenia, particularly since in younger patients, the "lean" nature of the phenotype may mean that important cardiometabolic investigations may be overlooked in the clinic.

## Strengths and limitations

Strengths of this study include the use of a large set of cardiometabolic traits and large GWAS datasets, through which we were able to test specific biological mechanisms. We chose SNPs reaching genome-wide significance from large GWAS and meta-GWAS for insulin resistance and related cardiometabolic traits. We performed a comprehensive set of sensitivity analyses to check MR assumptions. Furthermore, while weak instrument bias may be a factor in MR analysis, in two-sample MR, this bias tends towards the null [55], thus would not explain the positive associations we describe. We corrected for multiple testing to minimise potential type I error.

Our study has some limitations. We did not select SNPs in known coding regions for the exposures, for example, the *IRS-1* gene for insulin resistance [56]. We took this step on the assumption that many mechanisms at play may not yet be fully understood. For example, while the heritability of cardiometabolic traits such as obesity is as high as 70%, the variance currently explained by known genetic variants is but a small fraction of this [57]. In addition, selecting SNPs from many different GWAS studies featuring large sample sizes may increase the risk of sample overlap between exposure and outcome variables and can bias the results in either direction, depending on the proportion of overlap [27]. Also, for our primary inflammation-related SNP analysis, we chose a stringent *p*-value threshold to define inflammation-related SNPs. In doing so, we may have overlooked some SNPs with true inflammatory associations. As a result, only one genome-wide significant inflammation-related genetic variant was included in the analysis of fasting insulin and HDL, and none were included for triglycerides. Therefore, these results be considered with caution. However, we attempted to address this limitation by relaxing the *p*-value threshold for inflammation-related SNPs, thereby allowing a greater number of SNPs to be included, and the results for fasting insulin, HDL and triglycerides were consistent. Yet, the inclusion of inflammation-related genetic variants at a relaxed significance threshold may have increased the risk of weak instrument bias for those analyses. In the future, larger and better-powered GWAS may identify more SNPs for analysis and at greater resolution, potentially unearthing a larger number of inflammation-related SNPs, which would be helpful to confirm our findings. Additionally, the full range of gene products from the genetic variants we used as proxies for the cardiometabolic traits is unknown, and so we are unable to comment on potential biological mechanisms of association other than inflammation, which may also be relevant. Finally, our analyses were based on data from mostly European participants, so it is unclear whether our results apply to other populations. Large-scale GWAS and replication of our analyses in other populations are required to answer this question.

## Conclusions

It is well established that certain antipsychotic drugs and lifestyle factors such as smoking, lack of exercise, and poor diet are important contributors to cardiometabolic comorbidity in people with schizophrenia. In addition to this, our findings suggest that inflammation may be a

common cause for schizophrenia and cardiometabolic disorders, which may at least partly explain why they so commonly co-occur in clinical practice. Lifestyle modification and careful prescription of certain antipsychotic medications remain crucial malleable targets to reduce the significant impact that comorbid cardiometabolic disorders place on the quality and length of life in people with schizophrenia. In addition, our findings suggest that targeting inflammation could be an important therapeutic target for the treatment and prevention of cardiometabolic disorders in people with schizophrenia. Future research should seek to examine the biological mechanisms, which underpin how inflammation can simultaneously increase the risk of both insulin resistance and schizophrenia.

## Supporting information

**S1 Methods. Directed acyclic graphs outlining potential mechanisms of association between inflammation, insulin resistance, and schizophrenia.**
(DOCX)

**S2 Methods. GWAS used for SNP selection.**
(DOCX)

**S3 Methods. SNPs used as instruments for fasting insulin, triglycerides, and high-density lipoprotein.**
(DOCX)

**S4 Methods. SNPs used as instruments for fasting plasma glucose.**
(DOCX)

**S5 Methods. SNPs used as instruments for type 2 diabetes mellitus.**
(DOCX)

**S6 Methods. SNPs used as instruments for body mass index.**
(DOCX)

**S7 Methods. SNPs used as instruments for glucose tolerance.**
(DOCX)

**S8 Methods. SNPs used as instruments for low density lipoprotein.**
(DOCX)

**S9 Methods. SNPs used as instruments for glycated haemoglobin.**
(DOCX)

**S10 Methods. SNPs used as instruments for leptin.**
(DOCX)

**S11 Methods. MR analysis methods.**
(DOCX)

**S12 Methods. Inflammation-related SNPs for fasting insulin, triglycerides, and high-density lipoprotein.**
(DOCX)

**S13 Methods. Inflammation-related SNPs for low density lipoprotein.**
(DOCX)

**S14 Methods. Inflammation-related SNPs for fasting plasma glucose.**
(DOCX)

**S15 Methods. Inflammation-related SNPs for glycated haemoglobin.**
(DOCX)

**S16 Methods. Inflammation-related SNPs for type 2 diabetes mellitus.**
(DOCX)

**S17 Methods. Inflammation-related SNPs for body mass index.**
(DOCX)

**S18 Methods. Inflammation-related SNPs for schizophrenia.**
(DOCX)

**S19 Methods. SNPs used for CRP in MVMR analysis.**
(DOCX)

**S1 Results. Multivariable MR (MVMR) results for insulin resistance phenotype exposures (all-SNP analysis) with addition of CRP as exposure.**
(DOCX)

**S2 Results. Multivariable MR (MVMR) results for insulin resistance phenotype exposures (inflammation-related-SNP analysis) with addition of CRP as exposure.**
(DOCX)

**S3 Results. MR analyses using all SNPs for schizophrenia and cardiometabolic outcomes.**
(DOCX)

**S4 Results. The association between inflammation-related schizophrenia SNPs and cardio-metabolic outcomes.**
(DOCX)

**S5 Results. Cochran Q tests for heterogeneity and MR–Egger intercept tests for horizontal pleiotropy for the association between all cardiometabolic SNPs and schizophrenia.**
(DOCX)

**S6 Results. Cochran Q tests for heterogeneity and MR–Egger intercept tests for horizontal pleiotropy for the association between inflammation-related cardiometabolic SNPs and schizophrenia.**
(DOCX)

**S7 Results. Cochran Q tests for heterogeneity and MR–Egger intercept tests for horizontal pleiotropy for the association between schizophrenia SNPs and cardiometabolic outcomes.**
(DOCX)

**S8 Results. Cochran Q tests for heterogeneity and MR–Egger intercept tests for horizontal pleiotropy for the association between inflammation-related schizophrenia SNPs and car-diometabolic outcomes.**
(DOCX)

**S9 Results. MR-PRESSO tests of cardiometabolic all-SNP analysis to examine for and cor-rect horizontal pleiotropy.**
(DOCX)

**S10 Results. MR-PRESSO tests of inflammation-related cardiometabolic SNPs to examine for and correct horizontal pleiotropy.**
(DOCX)

**S11 Results. MR-PRESSO tests of schizophrenia all-SNP analysis to examine for and correct horizontal pleiotropy.**
(DOCX)

**S12 Results. MR-PRESSO tests of inflammation-related schizophrenia SNP analysis to examine for and correct horizontal pleiotropy.**
(DOCX)

**S13 Results. $I_{2GX}$ statistics to examine for potential violation of the "No Measurement Error" (NOME) assumption for MR–Egger analyses.**
(DOCX)

**S1 Checklist. STROBE-MR: Guidelines for strengthening the reporting of mendelian randomization studies.**
(DOCX)

**S2 Checklist. STROBE: Guidelines for reporting observational studies.**
(DOCX)

**S1 Fig. Forest plot illustrating MR analyses of schizophrenia as outcome using all SNPs (green) and inflammation-related SNPs (purple).**
(DOCX)

## Acknowledgments

The authors wish to thank Dr Isobel Stewart (University of Cambridge) for her methodological advice and support.

## Author Contributions

**Conceptualization:** Benjamin I. Perry, Stephen Burgess, Rachel Upthegrove, Claudia Langenberg, Nicholas J. Wareham, Peter B. Jones, Golam M. Khandaker.

**Data curation:** Benjamin I. Perry.

**Formal analysis:** Benjamin I. Perry, Hannah J. Jones, Amy M. Mason.

**Funding acquisition:** Benjamin I. Perry.

**Investigation:** Benjamin I. Perry, Stan Zammit, Rachel Upthegrove, Amy M. Mason.

**Methodology:** Benjamin I. Perry, Stephen Burgess, Hannah J. Jones, Stan Zammit, Rachel Upthegrove, Amy M. Mason, Felix R. Day, Claudia Langenberg, Nicholas J. Wareham, Peter B. Jones, Golam M. Khandaker.

**Resources:** Golam M. Khandaker.

**Supervision:** Stephen Burgess, Hannah J. Jones, Stan Zammit, Rachel Upthegrove, Felix R. Day, Nicholas J. Wareham, Peter B. Jones, Golam M. Khandaker.

**Visualization:** Benjamin I. Perry.

**Writing – original draft:** Benjamin I. Perry.

**Writing – review & editing:** Benjamin I. Perry, Stephen Burgess, Hannah J. Jones, Stan Zammit, Rachel Upthegrove, Amy M. Mason, Felix R. Day, Claudia Langenberg, Nicholas J. Wareham, Peter B. Jones, Golam M. Khandaker.

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
