## [Decision Letter · Decision Letter 0]

29 Jul 2020

Dear Dr. Perry,

Thank you very much for submitting your manuscript "Evidence for Inflammation as a Putative Shared Mechanism for Insulin Resistance and Schizophrenia: A Mendelian Randomization Study" (PMEDICINE-D-19-04114) for consideration at PLOS Medicine. 

Your paper was evaluated by a senior editor and discussed among all the editors here. It was also discussed with an academic editor with relevant expertise, and sent to three independent reviewers, including a statistical reviewer. The reviews are appended at the bottom of this email and any accompanying reviewer attachments can be seen via the link below:

[LINK]

In light of these reviews, I am afraid that we will not be able to accept the manuscript for publication in the journal in its current form, but we would like to consider a revised version that addresses the reviewers' and editors' comments. Obviously we cannot make any decision about publication until we have seen the revised manuscript and your response, and we plan to seek re-review by one or more of the reviewers. 

We expect to receive your revised manuscript by Aug 19 2020 11:59PM. Please email us (plosmedicine@plos.org) if you have any questions or concerns.

We look forward to receiving your revised manuscript. 

Sincerely,

Caitlin Moyer, Ph.D.

Associate Editor 

PLOS Medicine

plosmedicine.org

1. Please completely address the comments of reviewers 2 and 3, including clarification of the hypothesis put forth in the introduction, in particular regarding what is meant for inflammation being a “shared mechanism” behind cardiometabolic disorders and schizophrenia. As requested by reviewers, please also include more of an explanation of the rationale and implications behind the different MR methods, with more clarity in the interpretation of the results of the different tests performed, as this is helpful to readers who may lack specific knowledge of the MR methods.

2.Competing Interests: Please add this statement to the manuscript's Competing Interests: "SB is a paid statistical consultant on PLOS Medicine's statistical board." Please add this statement to the manuscript's Competing Interests: "CL is an Academic Editor on PLOS Medicine's editorial board."

3. Prospective protocol: Did your study have a prospective protocol or analysis plan? Please state this (either way) early in the Methods section.

4. Data Availability Statement: The Data Availability Statement (DAS) requires revision. You mention the sources of the summary statistics used in the manuscript. For each data source used in your study please note here: 

5. Abstract: Background: The final sentence should state the goal or question of the study.

6. Abstract: Methods and Findings: For the “large genome-wide association studies” please provide some detail describing the demographics of the participants of those studies, the names of the studies/cohorts.

7. Abstract: Methods and Findings: Line 21-22, and throughout: please reduce the use of causal language- we suggest: “Our study found no evidence in support of a causal relationship between schizophrenia and insulin resistance.” or similar. Please similarly adjust causal language throughout the manuscript.

8. Abstract: Please report effects with p values and 95% CIs when describing the main results of the study (associations between factors and schizophrenia individually and the multivariable MR findings with CRP).

9. Abstract: Methods and Findings: In the last sentence of the Abstract Methods and Findings section, please describe the main limitation(s) of the study's methodology.

10. Abstract: Conclusions: Please address the study implications without overreaching what can be concluded from the data; how do the results described in the Methods and Findings speak to the mechanism where treating inflammation may prevent cardiometabolic disease in schizophrenia?

11. Author Summary: At this stage, we ask that you include a short, non-technical Author Summary of your research to make findings accessible to a wide audience that includes both scientists and non-scientists. The Author Summary should immediately follow the Abstract in your revised manuscript. This text is subject to editorial change and should be distinct from the scientific abstract. Please see our author guidelines for more information: https://journals.plos.org/plosmedicine/s/revising-your-manuscript#loc-author-summary

12. Methods: Please provide the name(s) of the institutional review board(s) that provided ethical approval.

13. Methods: Please provide summary demographic information from the GWAS cohorts (e.g. population, setting, relevant characteristics).

14. Results: First paragraph: “The strength of evidence for these associations weakened substantially after correction for multiple testing” Please provide the OR, 95% CIs and p values for the findings corrected for multiple testing here. Similarly, in the paragraph describing the findings for IR-related SNPs, please provide the results when corrected for multiple testing.

15. Results: Top of page 17: “Test for Bidirectionality 1 using Schizophrenia as Exposure”: In this section, please revise to say, “We did not find a statistically significant MR association between…” or similar language throughout this paragraph. Similarly, please revise the first sentence of the next section “Test for horizontal pleiotropy”

16. Discussion: Second sentence of first paragraph: Please revise this sentence, and similar sentences throughout the text to make it clear that such associations did not reach the significance level that was adjusted for multiple comparisons, in addition to being inconsistent across MR methods. “In analyses using all associated SNPs, we report some evidence that genetically predicted levels of triglycerides, HDL and leptin may be associated with schizophrenia, although estimates were inconsistent across MR methods and may have been affected by horizontal pleiotropy.”

17. References: Please be sure to use the "Vancouver" style for reference formatting (including in the supplementary files), and see our website for other reference guidelines: https://journals.plos.org/plosmedicine/s/submission-guidelines#loc-references

18. Table 1 and Table 2: Please define all abbreviations found within the tables in the legends.

19. Figure 1: In the legend, please define abbreviations for IVW MR, SNPs, HDL, and please describe the nature of the points, whiskers and lines in the plots. Please increase the size of the fonts used.

20. Figure 2: Please define abbreviations for SNP in the legend.

21. Figure 3: Please define abbreviations for MVMR, and CRP, in the legend.

22. Supporting information tables and figures: Please provide legends for each individual table and figure in the Supporting Information. In the legends, please define each abbreviation that appears in the corresponding figure/table. Within the manuscript please refer to individual items as “S13 Table” to clarify to which supporting information item you are referring.

23. Checklist: Please ensure that the study is reported according to the STROBE guideline, and include the completed STROBE checklist as Supporting Information. When completing the checklist, please use section and paragraph numbers, rather than page numbers. Please add the following statement, or similar, to the Methods: "This study is reported as per the Strengthening the Reporting of Observational Studies in Epidemiology (STROBE) guideline (S1 Checklist)."

Comments from the reviewers:

Reviewer #1: Thank you for the opportunity to review this paper. It was a pleasure to read. The authors did an excellent job in presenting the background and rationale for the study as well as a complex series of analyses is a succinct fashion. The results are novel and important both for clinicians and researchers 

Reviewer #2: The aim of this study is to investigate whether insulin resistance is a causal risk factor for schizophrenia using publicly available summary genetic data and a two-sample Mendelian randomization (MR) design. Further to this, a secondary objective of the paper is to study if inflammation is a potential mechanism explaining this association. 

The research question is interesting and two-sample MR is emerging as a gold standard approach (despite it is not without limitations) to ask causal questions. However, I have a number of concerns regarding this study, mainly about how the authors appraised the existing evidence and interpreted their results, and the methods used to explore the question under study. Below, I highlight a number of major and minor concerns. 

Major concerns

1. A more careful critical appraisal of the literature is required throughout the manuscript. 

a. Page 4, lines 13-19. The evidence presented here is largely cross-sectional, but the authors appear to draw causal inferences from it ("therefore, IR, a significant risk factors for T2DM and obesity, and schizophrenia, may share pathophysiologic mechanism"). The authors should explicitly mention the risk of reverse causality here. Also, they reference a longitudinal study ((Perry et al. 2018)) which finds no evidence of a longitudinal association of IR and psychotic experiences, thus strengthening evidence of reverse causality. 

b. Reverse causality is mentioned on page 4, lines 21-23 but in relation to 'observational evidence', here it should be made clear that this statement refers to cross-sectional observational evidence, not all observational evidence. 

c. Page 4, lines 23-25 the authors should make it clear that residual confounding represents a separate issue that affects cross-sectional and longitudinal evidence. 

d. Overall, it would benefit the introduction to more clearly explain why in the absence of longitudinal evidence on the association between IR and schizophrenia, and the authors' previous evidence that IR does not mediate IL6/psychotic experiences findings, a causal role of the former and a mechanistic role of inflammation was hypothesized. 

e. The study cited on page 20 line 23 (reference 44) is based on 22 cases of schizophrenia and the study did not adjust for stressful life events, existing mental and physical illness and other important stressors that could confound the association between CPR and schizophrenia. These limitations should be noted, as at the moment the article is presented uncritically as providing evidence for CRP as a risk factor for schizophrenia. 

2. Methodological concerns: 

a. In lines 9-14 on page 5, the authors say that both CRP and IL6 have been shown to be associated with schizophrenia in both cohort studies, and MR analyses. However, evidence for the association between CRP and psychotic experiences from observational studies is a lot more tenuous than that for IL6. Similarly, in MR analyses lower CRP and higher IL6 are associated with schizophrenia (mentioned in the discussion). The authors should discuss this more clearly and should more explicitly explain the rationale for using CRP as opposed to IL6 in these analyses in light of this evidence. 

b. Lines 23-24 page 5, the authors say that they investigated whether inflammation could be a shared mechanism between IR and schizophrenia. However, it is not clear what is meant by that, e.g. does inflammation predispose to both IR and SCZ? Or does IR lead to inflammation which then leads to SCZ? More clarity around the hypothesized mechanisms is needed. 

c. In methods on page 8 'sensitivity analyses' the authors say that they repeat the analyses using only the inflammation-related traits in order to understand whether inflammation is a related mechanism. I am concerned that this, however, would simply capture the effect of inflammation on schizophrenia (more on this in point 3c below). The rationale for and methods used in these analyses should be explained more clearly.

d. I am not clear as to how the multivariable MR was conducted. It is my understanding, that in these analyses one should include all SNPs from both traits, using the shared SNPs as instrument in order to derive direct effect of each exposure. However, here this does not seem to have been done (or, if it has, it is not clear)? I would have expected to see the univariable coefficients for CRP and IR, as well as the mutually adjusted ones. 

e. I am not sure I follow the results presented in Figure 3. Here, the authors show the main analyses adjusted for CRP. These don't change much, which is to be expected as there is no overall association and CRP has a negative association with Schizophrenia (see:(Hartwig et al. 2017)). However, the authors also present the inflammation-related SNPs adjusted for CRP, which - on the other hand - show a reduction in association. I wonder if this is not to be expected though as CRP will be correlated with other inflammatory markers and CRP has a negative association with SCZ. I think the most interesting finding in figure 3 is the first two lines of each trait showing an overall lack of association. 

3. Interpretation of results: 

a. In the discussion, the authors say that their study shows some evidence of an association between triglycerides, HDL, and leptin and SCZ. However, there seems to be little to no evidence of an association with these markers based on the data presented in table 1 (particularly after correction for multiple testing). 

b. On page 19, lines 11-12, the authors repeat that they wanted to test whether inflammation was a mechanism linking IR to schizophrenia. However, it is not clear how this can be hypothesized in the absence of a main effect between IR and schizophrenia. 

c. On page 19, the authors say that their "results are consistent with the idea that IR may be associated with schizophrenia over and above the effect of confounding from socio-demographic, lifestyle, and other factors and that inflammation might be a common mechanisms". I am not sure how this inference was made, as there is no evidence of an association between IR and SCZ in the analyses. Showing an association between inflammation-related SNPs and SCZ only seems to validate the hypothesis of an inflammation (likely IL6 or other cytokines) and SCZ association? 

Minor concerns: 

4. On page 7, the authors explain that they use different MR methods depending on number of SNPs available. It would be helpful if they could add next to each of them the exposure they refer to. For instance, when they say "when <2 SNPs available" which traits are these? From the tables in the following pages I can't find any trait that only has one SNP. 

5. Please note that the sentence on page 7 lines 11-12 is incorrect. In a logistic regression, it is the log odds of the outcomes that are modeled, not the exposure. Also, similar to my previous comment, it would be helpful if the authors provided, in brackets, which exposures they are referring to, as this is not clear. 

References 

Hartwig FP, Borges MC, Horta BL, Bowden J, Davey Smith G (2017). Inflammatory biomarkers and risk of schizophrenia: A 2-sample mendelian randomization study. JAMA Psychiatry

Perry BI, Upthegrove R, Thompson A, Marwaha S, Zammit S, Singh SP, Khandaker G (2018). Dysglycaemia, Inflammation and Psychosis: Findings From the UK ALSPAC Birth Cohort. Schizophrenia Bulletin

Reviewer #3: This paper aims to understand causal relationships between inflammation, cardiometabolic risk factors (especially the insulin resistance phenotype) and schizophrenia, which can have potentially important clinical values. The way they select SNPs to run MR is interesting, my main concern is in how the authors can properly and more clearly interpret their results using different MR methods. Here are my comments in details:

1. From the MR analyses that have been done in the paper, do the authors suggest the following causal mechanism where inflammation causally affect both schizophrenia and an IR phenotype while the latter two are not causally related themselves? If so, this seems to be a very interesting hypothesis, and I would hope the authors to more clearly describe how the MR analyses done in the paper indicate/suggest it. If not, then I'm wondering why the authors chose to use only inflammatory-related cariometabolic SNPs in the second set of analyses. From the paragraph in page 8, seems that the authors want to do detect "vertical pleiotropy", but it is unclear to me what vertical pleiotropy and for what trait the authors want to detect, and why using the inflammatory-related SNPs can achieve their goal. 

2. The authors used the word "association" and "no evidence of association" to describe their results in page 10 and 12, which I think is inappropriate. MR methods are for detecting causal relationships between trait, not just association. If the MR p-values are not significant, it does not mean "no evidence of association".

3. I did not see an MR analysis using Schizophrenia as the exposure, is it possible that Schizophrenia is a cause for the cardiometabolic risk factors?

4. From Table 2, there are very few inflammatory-related SNPs used. Looking into Figure 1, some significant results (such as for HDL) seem to be due to only one SNP. Since there are so few SNPs, the MVMR method may not have enough power to detect any effect after adding CRP. I think the authors may want to state the limitation.

5. Related to 1, I'm wondering if showing CRP can be a cause of both schizophrenia and an IR phenotype using MR may strengthen the evidence that inflammation is the common mechanism.

6. In Table 1, are theses number of SNPs the numbers after LD clumping? I'm guessing yes while wondering why the "53" number of SNPs for fasting insulin, TG and HDL keep the same after clumping.

[LINK]

---

## [Decision Letter · Decision Letter 1]

15 Oct 2020

Dear Dr. Perry,

Thank you very much for re-submitting your manuscript "Evidence for Inflammation as a Putative Shared Mechanism for Insulin Resistance and Schizophrenia: A Bi-Directional Two-Sample Mendelian Randomization Study" (PMEDICINE-D-19-04114R1) for review by PLOS Medicine.

I have discussed the paper with my colleagues and the academic editor and it was also seen again by two reviewers. I am pleased to say that provided the remaining editorial and production issues are dealt with we are planning to accept the paper for publication in the journal.

[LINK]

We look forward to receiving the revised manuscript by Oct 22 2020 11:59PM. 

Sincerely,

Caitlin Moyer, Ph.D.

Associate Editor 

PLOS Medicine

plosmedicine.org

Requests from Editors:

1.Title: Please revise the title to: “The potential shared role of inflammation in insulin resistance and schizophrenia: A bi-directional two-sample Mendelian randomization study”

2.Data availability statement: Please remove the section describing data access from the body of the manuscript and update the Data Availability Statement in the manuscript submission system. This currently reads “Please see Data Availability Statement for information on downloading and accessing all publicly available data used in the manuscript.”

We suggest revising the text slightly for clarification: “The summary data for the 53 insulin resistance SNPs are available in the supporting information of Lotta et al [11] (10.1038/ng.3714). Full summary statistics for all traits used in the primary analysis are freely and publicly available for download at consortia/group websites. Specifically; for FI, FPG, HbA1C and glucose tolerance summary data, see https://www.magicinvestigators.org/downloads/; For HDL, LDL and triglycerides summary data, see http://csg.sph.umich.edu/willer/public/lipids2013/; For BMI summary data, see https://portals.broadinstitute.org/collaboration/giant/index.php/GIANT_consortium_data_file

s; For T2DM, see https://diagram-consortium.org/downloads.html; For leptin summary data, see

ftp://ftp.ebi.ac.uk/pub/databases/gwas/summary_statistics/KilpelainenTO_26833098_GCST0

03368; For schizophrenia summary data, see https://www.med.unc.edu/pgc/download-results/. Summary GWAS data for CRP, which formed part of our post-hoc sensitivity analysis, are also publicly available from the primary GWAS study [35], and inquiries regarding use of CRP summary data can be sent to s.ligthart@erasmusmc.nl.”

3.Abstract: Background, final sentence: Please avoid causal language here, we suggest revising to: “We aimed to examine whether there is genetic evidence that insulin resistance and seven related cardiometabolic traits may be causally associated with schizophrenia, and whether evidence supports inflammation as a common mechanism for cardiometabolic disorders and schizophrenia.” 

4.Abstract: Methods and findings: Please provide some broad demographic information on the individuals represented in the various studies- particularly the numbers of included individuals.

5.Abstract: Methods and Findings, top of page 3: Should “was associated with schizophrenia” be “were associated with schizophrenia”?

6.Abstract: Methods and Findings: Please revise to “We conclude that while our study did not find evidence in support of a causal relationship between schizophrenia and insulin resistance...” Also, this sentence could be moved to the “Conclusions” as this seems like more of an interpretation of the results.

7.Abstract: Conclusions: Please revise the first sentence to avoid causal language: “"Our findings support a role for inflammation as a common cause for...". Please also revise the final sentence, “...understand how inflammation may contribute to risk of schizophrenia and cardiometabolic disorders.”

8..Author summary: Why was this study done?: Please revise the second bullet point: “Insulin resistance, a precursor to diabetes, is sometimes detectable in young adults suffering…”

9.Author summary: Why was this study done?: There is a typo in the third bullet point: “...and so could be a common mechanism for schizophrenia…”

10.Author summary: What did the researchers do and find: Please revise the third and fourth bullet points as follows:

--After correction for multiple testing, overall, there was no significant evidence in support of a causal relationship between cardiometabolic traits and schizophrenia risk. However, we found evidence that supports a causal relationship between schizophrenia risk and an inflammation-related insulin resistance phenotype (comprising of raised fasting insulin, raised triglycerides and decreased high-density lipoprotein).

--Evidence for associations between schizophrenia risk and inflammation-related insulin resistance phenotypes attenuated fully in multi-variable MR analysis after adjusting for CRP, suggesting that these associations may be underpinned by inflammation.

11..Author summary: What do these findings mean?: Please combine and revise the first and second bullet points as follows:

--These results suggest that cardiometabolic traits are unlikely to have a causal role in the pathogenesis of schizophrenia, and vice versa. However, our results suggest that inflammation is related to the risk of both schizophrenia and cardiometabolic disorders, which may at least partly explain why they commonly occur in clinical practice.

Please revise the last two bullet points as follows: 

--Treating or preventing inflammation may be a putative therapeutic option for prevention and/or treatment of both schizophrenia and comorbid cardiometabolic disorders.

--In the future, more research is needed to understand the biological mechanisms underpinning how inflammation may increase the risk of schizophrenia and cardiometabolic disorders.

12. Introduction Page 6: Background should be renamed “Introduction”

13. Introduction: Page 7: Please revise to “...we have conducted a study to examine evidence in support of causality and shared mechanisms by testing the associations between IR and related cardiometabolic traits with schizophrenia.”

14. Methods: Page 9: “Our analysis plan was prospectively conceived by the authors in 2019” Please include the analysis plan as a supporting information file. If a documented plan is not available to include, please mention that the all described analyses were planned, or mention which were planned and when/why any data-driven changes to analyses took place, including changes made in response to peer review comments.

15. Results: Page 12, first and second sentence: Please revise to “We did not find significant associations between genetically-predicted levels of triglycerides and HDL with schizophrenia…” and “We also observed an apparent association between genetically-predicted leptin levels and schizophrenia, but this was inconsistent across MR methods and the evidence did not survive correction for multiple testing”

16. Discussion: First paragraph, first sentence: Please revise to reduce causal language: “...to first examine whether there are associations that support a causal relationship between IR and related cardiometabolic traits and schizophrenia, and second, to examine whether there is evidence supporting that inflammation may be a shared causal mechanism for IR and schizophrenia.”

17. Discussion: First paragraph, second sentence: Please revise to: “We report that the evidence in support of a causal association between genetically-predicted levels of triglycerides, HDL and leptin and schizophrenia was weak, in that the evidence of these associations did not survive correction for multiple testing, estimates were inconsistent across MR methods, and may have been affected by horizontal pleiotropy.”

18. Discussion: Page 19-20: Please revise to “In those analyses, we found strong and consistent evidence supporting that inflammation-related IR may have a causal relationship with schizophrenia.”

19. Discussion: Page 20: Please revise, as this is perhaps a likely interpretation but it cannot be said to be confirmed to be true: “This result suggests that the observed associations for the inflammatory-related variants are at least in part explained by inflammation.”

20. Discussion: Page 21: Please revise to: “The implications of our findings with regard to shared causal mechanisms should not distract clinicians from focusing on the assessment and management of malleable lifestyle factors related to cardiometabolic disorders in people with schizophrenia.”

21. Discussion: Page 23-24: Please edit this sentence to read: “In the future, larger and better-powered GWAS…”

22. Reference list: Please double check the formatting for all references, but particularly Ref 28, 36, and please see our website for reference guidelines https://journals.plos.org/plosmedicine/s/submission-guidelines#loc-references

23. Checklist: Thank you for including the STROBE-MR checklist and we agree this is an appropriate checklist for your study. However, it seems like the STROBE-MR is still under development/in the preprint stage (https://peerj.com/preprints/27857v1/) and for that reason please also include the STROBE checklist.

24. Supporting information file/results: It would be helpful to separate the supporting information items, particularly the results tables, into separate files.

25. S13 Results: Please spell out the abbreviation “NOME” in the title or legend.

26. S1 Figure: Please include a list of the beta coefficients with 95% CIs and p values unless they are already reported in another table.

Comments from Reviewers:

Reviewer #2: The authors have addressed most comments . I still have a couple of suggestions concerning the causal hypotheses in the introduction, as I don't think these come across very clearly at the moment. 

On page 7 the authors say "Inflammation could be a shared mechanism for cardiometabolic disorders and schizophrenia." Usually the term mechanism is used to describe causal pathways, whereas here it seems to be used to refer to the concept of confounding (i.e. inflammation causing both IR and SCZ). However, the authors then proceed to explain that they are testing whether IR is causally associated with SCZ (page 6) and whether "inflammation could be a shared mechanism linking IR and schizophrenia". If Inflammation is a common cause (i.e.: they hypothesise there is no effect of IR on SCZ after accounting for it), then IR cannot be said to be 'causally' associated with SCZ. As a result, at the moment it is still quite unclear what the causal hypotheses of these analyses are. Perhaps the authors should more clearly mention that they are testing 4 competing hypotheses and refer to the figure included in the revision already in the introduction. 

Related to the DAGs, I believe that in Panel A and Panel C there should also be an arrow between IR and SCZ (panel A) and IR and inflammation (panel C), as these describe confounding (e.g.: the authors test the association between IR and SCZ, but then test the role of inflammation as confounder)? There should be one in the other two as well, as I think it cannot be hypothesised prior to testing that indirect effect=total effect (i.e. there is no direct effect, once including the mediator).

Reviewer #3: The authors have addressed all my earlier comments.

[LINK]

---

## [Editor Report · Decision Letter 2]

26 Oct 2020

Dear Dr Perry, 

On behalf of my colleagues and the academic editor, Dr. Minelli, I am delighted to inform you that your manuscript entitled "The Potential Shared Role of Inflammation in Insulin Resistance and Schizophrenia: A Bi-Directional Two-Sample Mendelian Randomization Study" (PMEDICINE-D-19-04114R2) has been accepted for publication in PLOS Medicine. 

PRODUCTION PROCESS

Before publication you will see the copyedited word document (within 5 business days) and a PDF proof shortly after that. The copyeditor will be in touch shortly before sending you the copyedited Word document. We will make some revisions at copyediting stage to conform to our general style, and for clarification. When you receive this version you should check and revise it very carefully, including figures, tables, references, and supporting information, because corrections at the next stage (proofs) will be strictly limited to (1) errors in author names or affiliations, (2) errors of scientific fact that would cause misunderstandings to readers, and (3) printer's (introduced) errors. Please return the copyedited file within 2 business days in order to ensure timely delivery of the PDF proof. 

If you are likely to be away when either this document or the proof is sent, please ensure we have contact information of a second person, as we will need you to respond quickly at each point. Given the disruptions resulting from the ongoing COVID-19 pandemic, there may be delays in the production process. We apologise in advance for any inconvenience caused and will do our best to minimize impact as far as possible.

PRESS

PROFILE INFORMATION

Thank you again for submitting the manuscript to PLOS Medicine. We look forward to publishing it. 

Best wishes, 

Caitlin Moyer, Ph.D.

Associate Editor 

PLOS Medicine

plosmedicine.org